# Systematic review of contemporary interventions for improving discharge support and transitions of care from the patient experience perspective

Tiago S. Jesus [1,2]*, Brocha Z. Stern [3], Dongwook Lee [4], Manrui Zhang[2], Jan Struhar[5], Allen W. Heinemann [6,7], Neil Jordan[8,9], Anne Deutsch[6,7,10]

1 Division of Occupational Therapy, School of Health and Rehabilitation Sciences, College of Medicine, The Ohio State University, Columbus, Ohio, United States of America, 2 Center for Education in Health Science, Institute for Public Health and Medicine, Northwestern University Feinberg School of Medicine, Chicago, Illinois, United States of America, 3 Department of Population Health Science and Policy, Institute for Healthcare Delivery Science, Icahn School of Medicine at Mount Sinai, New York, New York, United States of America, 4 Center for Child Development & Research, Sensory EL, ROK, Dept. of Physical Medicine and Rehabilitation Medicine, Korehab Clinic, Dubai, UAE, 5 Nerve, Muscle and Bone Innovation Center & Oncology Innovation Center, Shirley Ryan AbilityLab, Chicago, Illinois, United States of America, 6 Center for Rehabilitation Outcomes Research, Shirley Ryan AbilityLab, Chicago, Illinois, United States of America, 7 Department of Physical Medicine and Rehabilitation Medicine, Northwestern University Feinberg School of Medicine, Chicago, Illinois, United States of America, 8 Department of Psychiatry and Behavioral Sciences, Dept. of Preventive Medicine, Northwestern University Feinberg School of Medicine, Chicago, Illinois, United States of America, 9 Center of Innovation for Complex Chronic Healthcare, Hines VA Hospital, Hines, Illinois, United States of America, 10 Center for Health Care Outcomes, RTI International, Chicago, Illinois, United States of America

* tiago.jesus@osumc.edu

## Abstract

### Aim

To synthesize the impact of improvement interventions related to care coordination, discharge support and care transitions on patient experience measures.

### Method

Systematic review. Searches were completed in six scientific databases, five specialty journals, and through snowballing. Eligibility included studies published in English (2015–2023) focused on improving care coordination, discharge support, or transitional care assessed by standardized patient experience measures as a primary outcome. Two independent reviewers made eligibility decisions and performed quality appraisals.

### Results

Of 1240 papers initially screened, 16 were included. Seven studies focused on care coordination activities, including three randomized controlled trials [RCTs]. These studies used enhanced supports such as improvement coaching or tailoring for vulnerable populations within Patient-Centered Medical Homes or other primary care sites. Intervention effectiveness was mixed or neutral relative to standard or models of care or simpler supports (e.g.,

**Funding:** Tiago S. Jesus completed part of this work and Manrui Zhang the totally of this work under a grant from the National Institute on Disability, Independent Living, and Rehabilitation Research (NIDILRR; 90ARHF0003). NIDILRR is a center within the Administration for Community Living (ACL), U.S. Department of Health and Human Services (HHS). The contents of this publication do not necessarily represent the policy of NIDILRR, ACL, or HHS, and the reader should not assume endorsement by the U.S. federal government. Dr Jesus initiated this work with the Northwestern University's affiliation (under the grant support mentioned) and the work was completed with The Ohio State University's affiliation. The funders had no role in study design, data collection and analysis, decision to publish, or preparation of the manuscript.

**Competing interests:** The authors have declared that no competing interests exist.

improvement tool). Eight studies, including three RCTs, focused on enhanced discharge support, including patient education (e.g., *teach back*) and telephone follow-up; mixed or neutral results on the patient experience were also found and with more substantive risks of bias. One pragmatic trial on a transitional care intervention, using a navigator support, found significant changes only for the subset of uninsured patients and in one patient experience outcome, and had challenges with implementation fidelity.

## Conclusion

Enhanced supports for improving care coordination, discharge education, and post-discharge follow-up had mixed or neutral effectiveness for improving the patient experience with care, compared to standard care or simpler improvement approaches. There is a need to advance the body of evidence on how to improve the patient experience with discharge support and transitional approaches.

## Introduction

The *patient experience* with care is an integral component of healthcare quality and a construct used to assess the *person-centeredness* of healthcare delivery, informing quality-improvement (QI) activities [1–3]. Specifically, *patient experience* refers to how patients have experienced key aspects of healthcare delivery, such as physician communication, involvement in decision-making, getting timely appointments, or care coordination and discharge planning activities across settings to address care fragmentation issues [1].

As synthesized in several systematic reviews, better patient experiences with care have been associated with improved treatment adherence and improved patient and health system outcomes [4–7]. For example, a recent systematic review found that better scores on standardized patient experience measures were associated with greater self-reported physical and mental health, lower frequency and length of hospitalizations, and fewer emergency room visits [4]. Furthermore, improving the patient experience of care has been recognized as a QI aim in itself, under the goal of improving the person-centeredness of care [8, 9].

Many health systems now reimburse providers for the value of care, including patient experience scores. The US Agency for Healthcare Research and Quality (AHRQ) has developed the Consumer Assessment of Healthcare Providers and Systems (CAHPS) surveys and quality measures, which have been widely used to track, publicly report, and compare providers' patient experience performance [10]. Within selected pay-for-performance programs, the US Centers for Medicare & Medicaid Services uses CAHPS scores when calculating payment incentives [11]. Several other countries require providers to monitor and improve patient experience [12, 13]. Overall, healthcare systems, organizations, and frontline practitioners are subject to quality mandates, value-based incentives, or market pressures (e.g., customer loyalty) to develop systematic activities to improve their patient experience performance [14], including for care coordination and discharge support [15, 16].

Reducing care fragmentation, improving continuity of care, and preventing rehospitalizations can be achieved by several interventions. Examples include supported discharge (i.e., coordinated transfer of care from hospital to patient's home), case management programs, transitional care services, use of care coordinators, as well as models of care such as patient-centered medical homes (PCMHs) [17, 18]. Systematic reviews have addressed the

effectiveness of these health service interventions, and many have been shown to be effective for a set of health, quality of life, and health system outcomes [17–20]. For example, interventions to facilitate the transition from hospital to home when started in the hospital and continued in the community through telephone or other telehealth follow-up or scheduled home visits can be effective in reducing hospital readmission, especially when tailored to each patient [18, 19, 21]. However, we found no systematic review focused on the impact that care coordination, discharge support, or transitional care activities have on patient experiences with care. A recent overview of systematic reviews for care coordination interventions found patient experiences were rarely addressed or reviewed as outcomes of these interventions [22].

Here, we aim to synthesize the contemporary literature (2015–2023) describing the impact of care coordination, discharge support, and transitional care activities on patient experience as assessed by formal patient-reported experience measures.

## Materials and methods

### Design

We registered (PROSPERO: CRD42022358337) a systematic review protocol that focused on synthesizing the English-language contemporary evidence addressing health service interventions to improve patient-reported experiences with care. Here, we focus on the impact of interventions related to enhanced care coordination, discharge support, or transitional care activities on the patient experience. We use the Preferred Reporting Items for Systematic Reviews and Meta-Analyses (PIRSMA) guidance for the review report (see S1 Appendix for the checklist).

### Eligibility

We included controlled trials, longitudinal observational studies with controls, and pre-post designs with >30 participants. We excluded qualitative, cross-national studies, or any study not assessing the impact on patient-reported experience measures. Studies needed to have full texts available, focus on improving the patient experience as one of two primary outcomes measures, and report inferential statistics (e.g., p-values) about the impact of the intervention on a patient experience measure.

For patient experience outcomes, we included studies that used standardized, quantitative patient experience assessments, including validated surveys, their items or composite domains, or surveys that were externally collected and routinely used across providers, including for value-based reimbursement. We excluded studies that assessed patient experiences only with non-standardized, qualitative, or other non-validated instruments. Measures that were not patient-reported (e.g., observational) were excluded. We did not synthesize the impact of the interventions on measures other than patient-reported experience with care.

For participants, we included health systems, organizations, providers, networks, settings, or service units, including any health professionals or staff. We excluded health service interventions exclusively delivered by students or clinicians-in-training. Those providing patient experience feedback could be the patient, family/informal caregivers, or any of those as proxy respondents on the patient's behalf.

For context, we had no geographic restriction, but we only included English-language articles published in 2015 or thereafter, to reflect contemporary interventions responsive to the recent focus or incentives for improving the patient experience of care. For example, value-based reimbursements that included patient experience scores did not emerge in the US until 2013 [23].

For this review, we only included health-service improvement interventions focused on *care coordination* (i.e., systematic organizational, service line, or service unit strategies to address fragmentation of care delivery and enhance continuity of care [22]), *discharge support* (i.e., patient education in preparation for discharge and/or post-discharge education or follow-up support either in the environment or by telephone or other telehealth/web-based mechanisms [18]), and *transitional care* (i.e., subset of intermediate, time-limited, mediated set of actions to ensure the coordination or continuity of care as patients transfer between different settings or levels of care [24]), which often involves a single point of contact to optimize service access, communication, and coordination [25].

## Search

We searched six scientific databases (PubMed/MEDLINE, CINAHL, EconLit, PsycINFO, DOAJ, and Scopus) using a combination of free-text words with indexed terms that reflect our eligibility criteria. We restricted the search for English language and publications after January 2015. S2 Appendix shows the detailed searches for each database. Searches were initially conducted in December 2022 and updated in January 2024.

We also conducted targeted searches within the *Patient Experience Journal*, *Journal of Patient Experience*, *Medical Care*, and *Health Expectations*. The journal *NEJM Catalyst Innovations in Care Delivery* was searched since January 2020, when the journal became peer reviewed. These were journals with frequent papers identified in the initial database searches. Finally, we conducted snowballing strategies (e.g., citation-tracking, similar publications in PubMed) using the final included articles and reference lists of similar systematic reviews, such as those identified through the searches.

## Selection

Two independent reviewers (TJ first reviewer; DL, JS, or MZ for the second reviewer role) conducted the title-and-abstract screenings and then the full text reviews against the eligibility criteria. A third reviewer (AD) was engaged to resolve conflicts.

## Data extraction

The research team built a data extraction form that they used to chart data such as study characteristics and context, patient experience measures and metrics, interventions, analytic approaches, and results pertaining to patient-reported experience with care. The data were charted initially by the first author (TJ) and verified by a second author (BS), with subsequent rounds for any flagged extractions until they reached consensus. S3 Appendix provides detailed, consensus-based data extractions for each included study.

## Quality assessment

The quality assessment was performed independently by TJ and BS. The Cochrane-suggested risk of bias criteria for Effective Practice and Organization of Care (EPOC) reviews were used for the controlled trials [26], while the National Heart, Lung, and Blood Institute's quality assessment tools were used for the pre-post studies and observational studies that assessed or compared the effects of health service interventions [27].

As planned in the review protocol, we did not grade the strength of the evidence across the research studies and their design. Covering a range of health service or QI interventions, contexts, assessments, and study designs, our review is configurative (not aggregative) in nature. We tabulated the consensus-based appraised risks of bias for each study's context and design

based on the respective checklist-based methodological assessment; those detailed assessments are in S4 Appendix.

## Analytic plan

We tabulated and synthesized the results per three major inclusion categories: 1) care coordination, 2) discharge support, and 3) transitional care. Within each category, we tabulated the methods and appraised the risk of bias for each study, and then synthesized the interventions and their findings, ordered by study design (starting with RCTs) and publication date within the same design. Given the configurative nature of the review and the heterogeneity of interventions and study contexts, we did not conduct a meta-analysis.

## Results

Fig 1 shows that among the 1240 unique papers screened, 143 full texts were assessed for eligibility and 16 were included [15, 16, 28–40]. The absence of impact assessment with inferential statistics was the main reason for exclusion at the full-text assessment. Among papers included, seven focused on additional care coordination supports beyond those provided as standard in PCMHs; eight focused on enhanced discharge supports, specifically for enhanced patient education, reengineered discharge, telephone follow-up, or a mix of those interventions; and one focused on transitional care.

### a) Enhanced care coordination supports—Primary care level (*n* = 7)

Table 1 summarizes each study's methods (e.g., design, context) and risks of bias; Table 2 summarizes the intervention and study findings.

Context-wise, all studies were multi-site PCMHs or other primary care practices, all from a network or integrated health system in the US: Veterans' Health Administration (VHA) [34–36], a statewide PCMH network [37], Federally Qualified Health Centers [38], a Safety Net Medical Homes' Initiative [40], or academically-affiliated primary care practices [39].

Three studies were RCTs for facilitating site-based improvement projects. A recent RCT at the VHA, with the fewest appraised risks of bias (Table 1), used QI facilitation such as coaching by centralized improvement experts to support the site's use of a care-coordination improvement toolkit–beyond its internal dissemination (control); in both groups improvements in the patient experience were found, with no significant benefit for the more resource-intensive arm that used coaching facilitation [36].

Another RCT focused on the impact of augmented care coordination for patients with high risk of hospitalization (beyond standard PCMH; control group) and found improvements in 2 of 6 care coordination items assessed from the patient experience perspective [35]. The remaining RCT focused on encouraging the selection of high-value care-coordination QI goals (beyond standard QI facilitation: control group) and found mixed results [37]. These RCTs had appraised risks of bias (e.g., potential contamination [35]; selection bias [37]).

Among other study types, one non-randomized trial focused on nurses providing additional care coordination and navigation support for socially vulnerable patients, beyond a PCMH model (control), and found no statistically significant differences between groups ($p$ = 0.07) [38]; Finally, three observational studies (tailoring PCMH model to homeless patients [34]; practice transformation tailored to the safety nets [40]; learning collaborative [39]) showed mixed results and had substantive risks of bias (Table 1).

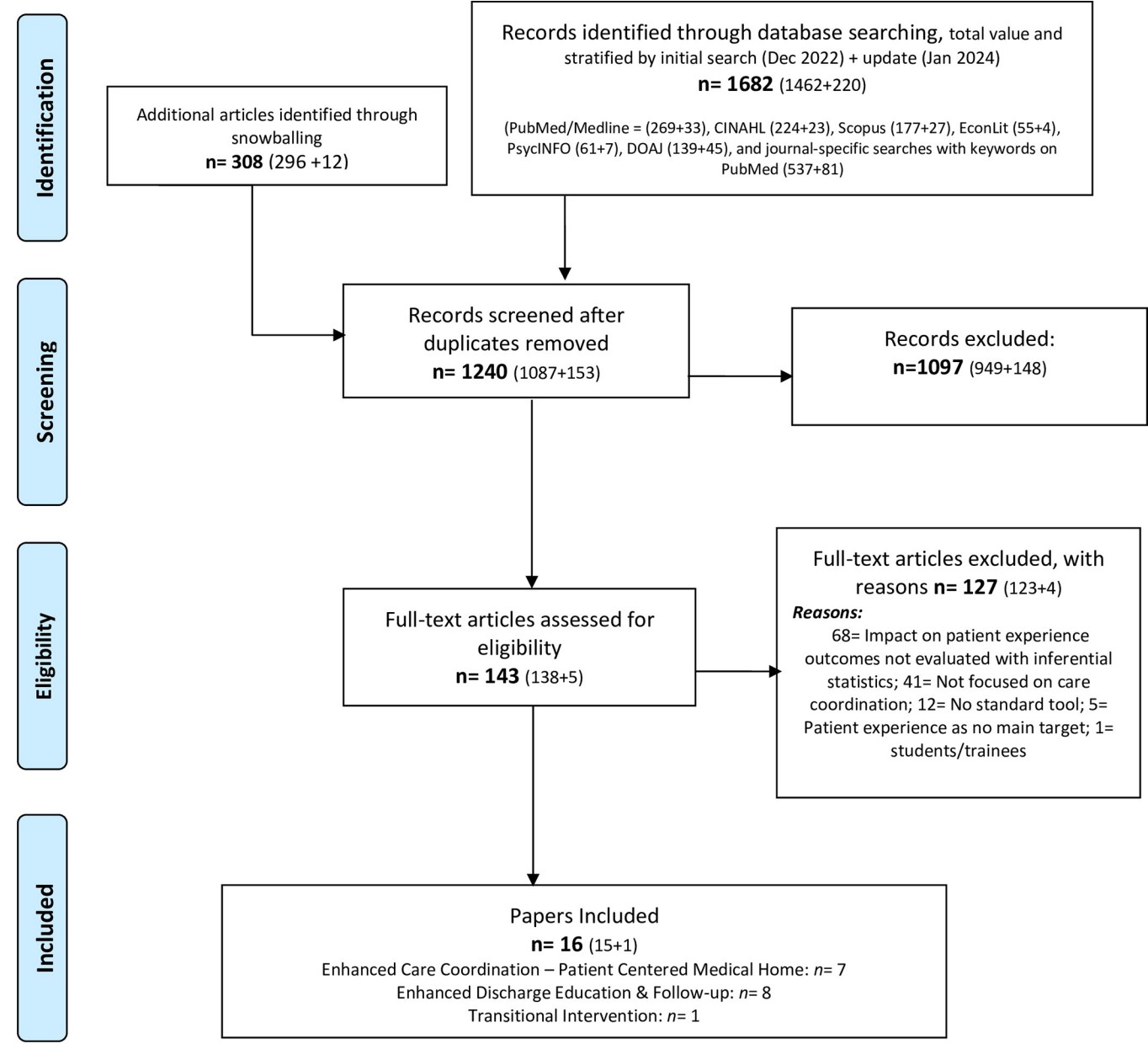

**Fig 1. PRISMA flowchart; results are stratified by the initial and updated searches.**

## b) Enhanced discharge or follow-up supports—After hospital or surgical care (*n* = 8)

Table 3 summarizes the methods and Table 4 the interventions and results for the studies on enhanced discharge support, including patient education or follow-up. All studies were based in the US.

Among the three RCTs, two focused on patient discharge education using the *teach back* method, after a health literacy assessment [28] or as part of a multi-modal language-concordant discharge support and telephone follow-up [29]. None of the study interventions led to better experience results than standard care. The other RCT was on a web-based, surgical

**Table 1. Summary of study methods and risk-of-bias assessment: Enhanced care coordination for Patient-Centered Medical Homes or other primary care practices.**

| | Design & Analysis | Context / Setting | Risk of Bias–level (within each study design) & Described risks | |
|---|---|---|---|---|
| Noël et al, 2022 [36] | RCT (cluster-randomized) of QI approaches. Multivariable difference-in-differences with adjustment for clustering and patient characteristics. | Patient-Centered Medical Homes, primary care; 12 sites (6 intervention) within the Veterans Health Administration system, USA | RCT | Different pre and post groups. Non-coached clinics had significantly lower baseline values for the outcome, thus greater improvement margins. |
| Zulman et al., 2019 [35] | Secondary survey of a RCT (patient-level randomization) for quality improvement. Logistic regression, using intention-to-treat analyses. Sensitivity analyses with adjustment. | Patient-Centered Medical Homes, primary care; 5 sites within the Veterans Health Administration system, USA | RCT | No baseline outcomes. Controls were from the same sites, with potential for contamination. Not all patients randomized to the intervention received the intervention. Patients may not distinguish the intervention from the previously provided support (control). Self-selected sites participated in the program, with 5% greater experience scores than the national average for several health care experience survey questions. No correction for multiple comparisons. |
| Dorr et al., 2016 [37] | Pragmatic RCT (cluster-randomized) for quality improvement. Adjusted difference-in-differences analysis and generalized estimating equation approach. | Patient-Centered Medical Homes, primary care; 8 sites (4 intervention, 4 control), Oregon, USA | RCT | Use of P value threshold of $P < .10$ and all significant changes identified ranged between $P > 0.5$ and $P < .10$. No correction for multiple comparisons. Different pre and post groups. Small number of clusters. Selection and respondent bias unable to be adjusted for, e.g., those motivated to take part in the experiment and respond in the pre- and post-periods. |
| Nembhard et al., 2020 [38] | Clustered, non-randomized multi-site controlled before-and-after study. Difference-in-differences analysis. | Patient-Centered Medical Homes (Federally Qualified Health Centers), primary care; 12 sites (6 intervention, 6 control) within a primary care network, USA | NR-CT | No randomization. Centers part of one network and from one state. Selection bias for participant sites. |
| Jones et al., 2019 [34] | Comparative, retrospective cohort study (one intervention group; two comparators). Multivariable logistic regression models (sensitivity analyses). | Patient-Centered Medical Homes, primary care; 510 sites (25 with intervention, 485 control) within the Veterans Health Administration system, USA | Obs-C | Observational, retrospective study. No baseline outcome measured. The outcome measure does not address homeless-specific concerns and service needs. Limited response rate. Site-specific inclusion criteria for the intervention. No correction for multiple comparisons. |
| Tung et al., 2018 [40] | Comparative repeated cross-sectional design with serial random samples. Adjusted difference-in-differences analysis and cross-sectional adjusted post-intervention generalized estimating equations. | Safety Net Medical Home Initiative's primary care practices: 24 sites randomly selected from 65 intervention sites (13 sites included in analyses), USA | Obs-C | Observational study. Baseline patient experience assessed 13 months after intervention began. All analyses at clinic level with inability to follow individual patients over time. Potential selection bias: only 13 of the 24 clinics that completed baseline surveys were included in the analyses as they also completed the post-intervention ones. No correction for multiple comparisons. |
| Nguyen et al., 2020 [39] | Comparative retrospective study. Adjusted differences-in-differences and regression models (inverse probability weighting in primary analysis; used sensitivity analyses). | Academically affiliated primary care practices; 50 sites (13 intervention, 37 comparison) affiliated with Harvard Medical School, USA | Obs-C | Observational study with external comparators with unknown practice patterns. There may have been similar practice changes in the comparison sites in line with general health care reform initiatives. Although adjusted models were used, the study does not control for variation in the size, structure, and composition of the care teams. No correction for multiple comparisons. |

RCT: Randomized Controlled Trial; NR-CT: Non-Randomized Controlled Trial; Obs-C: Observational, comparative study. QI: Quality Improvement.

**Table 2. Enhanced care coordination for Patient-Centered Medical Homes (PCMHs) or other primary care practices evaluated from the patient experience perspective as a primary outcome.**

| | Intervention type *Enhanced Care Coordination Supports*: | Design | HCSHS Score | SHEP [a] Top-Box | SHEP [a] Categorical | PACIC [b] Score | CSI Top Box | CSI Score | CG-CAHPS Score | PES Score | Results (Descriptive Synthesis) |
|---|---|---|---|---|---|---|---|---|---|---|---|
| Noël et al, 2022 [36] | **Enhanced QI facilitation & coaching** (beyond QI toolkit dissemination) | RCT | +x | | | | | | | | Both the intervention and control groups had significant pre-post care improvements in the patient experience of care coordination, but with no significant between-group difference over time (95% CI (− 0.47, 0.50))–i.e., no benefit for adding the more resource-intensive facilitating coaching component. |
| | Centralized, guided coaching of the participant VA primary-care clinics under a PCMH model. Coaching including a site visit, distance coaching, and cross-site learning on the use of a care coordination improvement toolkit (intervention) *vs.* internal toolkit dissemination alone (control) | | | | | | | | | | |
| Zulman et al., 2019 [35] | **Augmented care coordination activities for high-risk patients** (beyond standard PCMH) | RCT | x | + x | | + | + X | x | | | Care coordination outcomes: Intervention had better results than the controls for 2/6 SHEPS items: talked about health goals (73.1% *vs*. 68.4%; 95% CI = 1.00–1.59) and talked about barriers to taking care of health (60.4% vs. 54.8%; 95% CI = 1.02–1.56). PACIC chronic illness care scale: Intervention group had higher mean scores (2.91 vs. 2.75, *P* = .022). Trust outcomes: Intervention better than controls (60.5% *vs.* 53.1%; 95% *CI* = 1.10–1.66). Satisfaction outcomes: Intervention better than controls for satisfaction with primary care services (36.5% vs. 31.7%; 95% CI = 1.05–1.47) but not for other items or total score. Perceived access outcomes: no significant differences. |
| | Augmented care coordination activities for patients at high risk for hospitalization. Activities include home visits, health coaching, accompaniment to specialists, and intensive social work. These activities were delivered in addition to the standard PCMHs' support (control). | | | | | | | | | | |
| Dorr et al., 2016 [37] | **Encouraging selection of high-value care coordination QI goals** from a 12-item list (beyond standard QI) | RCT | | | | | | | + x | | The intervention performed better in 2 of 11 composite items, including "Follow-up on test results" (*p* = .091) and "Patients' rating of the provider" (*p* = .091). The control group performed better in "Access to care" (*p* = .093). |
| | Encouragement of PCMH sites (state of Oregon) for selecting QI goals from a list of 12 high value elements in addition to the standard QI facilitation (control): standard QI had IT-based milestone reporting, financial incentives on self-selected QI goals, and QI practice facilitator. | | | | | | | | | | |
| Nembhard et al., 2020 [38] | **Nurses take on added coordination & navigating support for high-risk patients**, i.e., socially vulnerable (beyond PCMH). | NR-CT | | | | | | | x | | Difference-in-differences analysis showed no significant difference in patient care experiences between the groups (*p* = 0.07). |
| | Nursing staff taking on additional coordination and navigation support roles for high-risk, socially vulnerable patients. Nurses also led weekly panel management meetings with enrollees' primary care and behavioral health providers. New role supported by training, a "playbook", and new electronic dashboard. Feedback reports provided to nurses on their performance. | | | | | | | | | | |

*(Continued)*

**Table 2.** (Continued)

| | | | HCSHS | SHEP [a] | PACIC [b] | CSI | CG-CAHPS | PES | |
|---|---|---|---|---|---|---|---|---|---|
| Jones et al., 2019 [34] | **Tailoring PCMH model to homeless patients** (beyond standard model) | Obs-C | | + x | | | | | In adjusted analyses, compared to same-facility controls, intervention group had better experiences in access ($p < .001$), communication ($p < .01$), office staff helpfulness/ courtesy ($p < .01$), provider ratings ($p < .05$), and comprehensiveness ($p < .05$). When compared to facilities without intervention, significant differences were found for communication and self-management support ($p < .05$). Care coordination and shared decision-making were not significantly different for the intervention group versus either comparator. |
| | Homeless-tailored medical home with tailored resources (above standard VA primary care—controls) to enhance access, address social determinants, and facilitate housing placement. For instance, providers are trained to deliver care for the homeless and further coordinate care with other services (e.g., mental health, addiction, and social). | | | | | | | | |
| Tung et al., 2018 [34] | **Practice transformation guided and tailored** to the safety net context | Obs-C | | | +x | | | | In the adjusted models, groups did not have significantly different total PACIC scores (95% $CI$: -1.1 to 16.5). However, at completion of the intervention, a 10-point higher PCMH capability score was associated with 8.9 points higher total PACIC score (95% $CI$: 3.1–14.7). Greater patient experience scores in favor of the high capability group were also observed in 4/5 subdomains (patient activation, delivery system design, contextual care, and follow-up/coordination). |
| | Practice transformation guided by eight *change concepts*, based on medical home principles tailored to the safety-net setting, supported locally by a Regional Coordinating Center with practice coaches. While all sites had the same intervention, the study compares high-improvement medical home transformation (high-capability settings) vs. low-improvement settings (controls). | | | | | | | | |
| Nguyen et al., 2020 [39] | **Learning collaborative** facilitating a primary-care practice transformation | Obs-C | | | | | | + X | Relative to comparison practices, the communication score in intervention practices increased by 1.47 percentage points on a 100-point scale ($P = .02$) between pre and post periods. No significant immediate improvements in the 5 other composite measures of patient experience of care. |
| | Learning collaborative for a primary care transformation initiative (multi-year, multi-site, phased, using lump sum payments) toward a team-based care. The collaborative included regular care team huddles, coaching, triannual 1.5-day learning sessions, and monthly webinars to discuss QI strategies. | | | | | | | | |

**Keys:** Results per measure & metric: + significant positive impact; **x** non-significant impact.

HCSHS: Health Care (System) Hassles Scale; SHEP: VA's Survey of Healthcare Experiences of Patients (adapted from CAHPS); PACIC: Patient Assessment of Chronic Illness Care; CSI: Consumer Satisfaction Index; CG-CAHPS: Consumer Assessment of Healthcare Providers and System Clinician & Group Survey; PES: Patient Experience Survey; RCT: Randomized Controlled Trial; NR-CT: Non-Randomized Controlled Trials; PPT: Pre-Post Test study; Obs-C: Observational, comparative study. QI: Quality Improvement. VA: Veterans Administration. IT: Information Technology.

[a] Jones et al. specifically reported use of the PCMH-SHEPS.

[b] Tung et al. reported use of a modified version of the PACIC.

patient education tool for pre-operative and follow-up instructions, and found improvements in the overall experience score [30]. However, several risks of bias were appraised in all the RCTs such single-center study [28–30], limited follow-up [29, 30], or no correction for multiple comparisons [28–30] (Table 3).

**Table 3. Summary of study methods and risk-of-bias assessment: Discharge education and follow-up or transitional support from an inpatient or surgical setting.**

| | Design & Analysis | Setting | Risk of Bias within each study design & described risks | |
|---|---|---|---|---|
| van Eck et al., 2018 [30] | Single-site RCT (patient-level randomization) with posttest measure only. T test and analysis of variance on covariates. | Private practice orthopedic clinic, outpatient and ambulatory surgery; Kerlan Jobe Orthopaedic Clinic, California, USA | RCT | Single site with 2 surgeons. Patients not masked to group allocation. No baseline outcomes. No correction for multiple comparisons. Limited follow-up time (only two weeks of post-assessment). |
| Griffey et al., 2015 [28] | Single-center RCT (patient-level randomization) with intention-to-treat analysis. Multivariable ordinal logistic regression. | Hospital, emergency department; Barnes Jewish Hospital, St. Louis, Missouri, USA | RCT | No baseline outcomes assessed. Single site. Convenience sampling prior to randomization. About 40% loss to follow-up. No random sequence generation for group assignment or allocation concealment. Contamination may have occurred. No correction for multiple comparisons. |
| Chan et al., 2015 [29] | RCT (patient-level randomization stratified by language). χ2 test on the top-box scores. Sensitivity and subgroup analyses. | Safety-net hospital, various units (internal medicine, family medicine, cardiology, neurology); San Francisco General Hospital and Trauma Center, California, USA | RCT | No baseline outcomes assessed. Single site. Only one month of post-intervention data. Medical teams not blinded to presence of intervention, and possible contamination across nurses within the hospital. No correction for multiple comparisons. |
| Centrella-Nigro & Alexander, 2017 [33] | Nonrandomized trial, two units of one hospital. Independent T tests. | Community Magnet-designated hospital, medical units (intervention unit mainly neurological patients, control unit mainly genitourinary and infectious diseases); Northern New Jersey, USA | NR-CT | No randomization. Clinical differences in units. Difference in baseline outcomes not adjusted for. Baseline characteristics not assessed or adjusted for. Potential for contamination because all units were in a single hospital. No correction for multiple comparisons. Lack of transparent reporting of participant numbers pre- and post-intervention. No combined analysis of between-group and between-time. |
| Cancino et al., 2017 [16] | Nonrandomized posttest design, two control groups with standard discharge (one in same unit and one in different unit). χ2 test on the top-box scores. | Safety-net hospital, adult inpatient family medicine (adult general medical units as one comparator); Boston Medical Center, Massachusetts, USA | NR-CT | No randomization. No baseline characteristics or outcomes data. No adjustment for covariates or correction for multiple comparisons. Low response rate. Possibility for contamination as some controls were within the same unit. Nurses selected patients (potential selection bias). Results not transferable to sites with higher baseline scores. |
| Schreiter et al., 2021 [15] | Historical cohort of matched controls at patient level. Wilcoxon rank-sum for scores and Fisher exact test for % of top-box scores. | Academic hospital, transition from surgery; University of Wisconsin Hospital, Wisconsin, USA | Obs-C | No randomization. Historical controls even though matched for some clinical indicators. Single site. No baseline outcomes assessed. Unadjusted analyses for outcome of patient experience. No correction for multiple comparisons. |
| Thum et al., 2022 [31] | Pre- and post-test, one organization with multiple units. χ2 on top-box scores and Kruskal–Wallis for the percentile rank change. | Academic tertiary care center and its affiliated community hospital; Thomas Jefferson University Hospital and affiliate, USA | PPT | No control group. Eligibility criteria not clearly described. Single pretest (although long preintervention period). No establishment of stable baseline through multiple measurement time points. No report on the numbers of surveys used for the analysis, only those mailed. No correction for multiple comparisons. |

*(Continued)*

**Table 3.** (Continued)

| | Design & Analysis | Setting | Risk of Bias within each study design & described risks | |
|---|---|---|---|---|
| March et al., 2022 [32] | Pre- and post- test, single-center retrospective review of a pilot program. χ2 test on the top-box score after T test on baseline characteristics. | Hospital, pharmacy; Methodist University Hospital, Memphis, Tennessee, USA | PPT | Retrospective, single-site pre-post-test analysis not adjusted for covariates. No establishment of stable baseline through multiple measurement time points. No correction for multiple comparisons. Intervention not delivered to all the eligible patients (business hours). No report on the number of patient experience surveys collected. Risks of selected findings: two different designs for two outcomes. |

RCT: Randomized Controlled Trial; NR-CT: Non-Randomized Controlled Trial; PPT: Pre-Post Test study; Obs-C: Observational, comparative study.

Two non-randomized trials were found: one on the implementation of *ReEngineered Discharge* for three patients a day that nurses selected as complex cases [16], the other on *teach back* discharge education by nurses [33]. Both achieved improvements in selected patient experience items such as in information provision. An observational study with historical controls focused on a telephone follow-up after hospital discharge with possible remediation plans or rescue resources (e.g., same-day clinic appointment) [15]. The intervention achieved higher percentages of top-box scores (i.e., highest score on the given patient experience items) compared to controls for 5 of the 11 patient experience items. Several risks of bias applied to the three studies (Table 3), including risks of contamination and unadjusted analyses.

Finally, two pre-post studies were included, one using *teach back* and other discharge redesigns [31], the other a pharmacist-led discharge education and follow-up intervention [32]. These provided either positive or mixed results (improvement in top box percentages but not in the percentile ranks with peer providers). In addition to the uncontrolled study design, these studies had multiple substantive risks of bias (Table 3), limiting the ability to draw conclusions.

### c) Transitional care intervention

One pragmatic clinical trial assessed the impact of a transitional care intervention from the hospital to the community, supported by community health workers and peer coaches [41]. Table 5 summarizes the methods, context, risk-of-bias assessment, intervention, and results. This navigator intervention was not effective at improving the patient experience as assessed by between-group differences for changes in informational support, although a subgroup analysis identified greater improvements for informational support among participants without health insurance. Suboptimal implementation fidelity and limitations of the measure used for the scope of the intervention (Table 5) may have contributed to the nonsignificant results.

### Discussion

This knowledge synthesis addresses a gap in the literature regarding the impact of care coordination, discharge support, and transition of care interventions on standardized patient experience measures. Health-service improvement activities, including enhanced supports or tailoring for vulnerable populations, achieved mixed to null improvements in the patient experience with the care coordination or discharge processes, compared to usual care or simpler quality-improvement supports. This was found by seven studies (including three multi-site

**Table 4. Discharge education and post-discharge follow-up (e.g., transitional care supports) after inpatient or surgical care evaluated from the patient experience perspective as a primary outcome.**

| | Intervention Type *Discharge education & post-discharge supports*: | Design | OAS-CAHPS Score | CAHPS Score | CTM-3 Score | HCAHPS Top-Box | HCAHPS Top-Box | HCAHPS Percentile | Pres Ganey Top-Box | Results—Descriptive Synthesis |
|---|---|---|---|---|---|---|---|---|---|---|
| van Eck et al., 2018 [30] | **Web-based, Surgical Patient Education Tool** for pre-operative & follow-up instructions | RCT | + x | | | | | | | Statistically significant higher OAS CAHPS total patient experience score for the intervention vs. control (97 ± 5 vs. 94 ± 8; *P* = .019). At the domain level, groups only significantly differed for "recovery" (92 ± 13 for intervention vs. 82 ± 23 for control; *P* = .001). No significant difference in total patient satisfaction score based on addition of video. |
| | The intervention group received access to an interactive web-based education tool in addition to routine perioperative instructions. The tool included instructions 14 days before and after the surgery, including preoperative instructions, day-of-surgery expectations, and postoperative precautions. A random subset of intervention patients also accessed links to videos demonstrating the surgery. | | | | | | | | | |
| Griffey et al., 2015 [28] | *Teach back* **for discharge instructions**, after health literacy assessment | RCT | | X | | | | | | No significant differences in the multivariable models for any the 4 items on the standard patient-reported experience measure (*P* values from 0.19 to 0.81), even though the intervention group had higher comprehension of medication (*P* < .02), self-care (*P* < .03), and follow-up instructions (*P* < .0001). |
| | *Teach back*, after a rapid health literacy assessment, applied to discharge instructions. Delivered by trained medical student Research Assistant (RA) alongside charge nurses who have had *teach back* education. Control group had usual discharge instructions. | | | | | | | | | |
| Chan et al., 2015 [29] | **Language-concordant discharge support, follow up and transitional support**, based on the ReEngineered Discharge, including *teach back* & motivational interviewing | RCT | | | x | x | x | | | No statistically significant differences between groups in CTM-3 scores (80.5% vs. 78.5%; p = 0.18) or HCAHPS discharge communication domain score (74.8% vs. 68.7%; *p* = 0.11), medicine communication (44.5% vs. 53.1%; *p* = 0.13), or nurse communication (67.9% vs. 64.9%; *p* = 0.43) scores. When stratified by language, no significant differences were seen. |
| | Language-concordant, nurse-led, hospital-based, tailored discharge support, follow-up, and care transition intervention (in addition to usual discharge planning as control group). Language-concordant inpatient nurse visits, motivational interviewing, *teach back*, discharge materials (a booklet in the patients' native language—conceptually like that used in ReEngineered Discharge (RED)), and phone follow-up that included, for example, a nurse organizing post-discharge services for participants or transportation plans for scheduled appointments, including instructions for arranging follow-up services, if necessary. | | | | | | | | | |
| Centrella-Nigro & Alexander, 2017 [33] | *Teach back* **for discharge instructions**, after 1-hour training of nurses | NR-CT | | + X | | | | | | Significant improvement in top box percentages for 1 of 7 tested items for the intervention group, on information about new medicine: 53.2% vs. 78.5%, p = .025. That item had a non-significant change for the control group: p = .932 |
| | *Teach-back* implementation, after 1-hour, multi-modal training of nurses. Regular audits of charts by nurse supervisors for assessing fidelity. | | | | | | | | | |
| Cancino et al., 2017 [16] | **Use of the ReEngineered Discharge (RED) Toolkit** for selected high-complexity cases | NR-CT | | | | | | | + x | Better top box percentages on the item about instructions: 61% *vs.* 35% same-unit controls (*p* < .001) *vs* 41% other-units controls (both *p* < .001). No significant difference on the item about feeling ready to be discharged: 45% *vs.* 35% same-unit controls (*P* = .15) *vs.* 51% other-units' controls (*p* = .25). |
| | Implementation of the ReEngineered Discharge (RED) Toolkit: 12-item patient education & discharge preparation processes. A trained nurse delivered the RED to about three patients a day, selected based on high-complexity discharge planning or history of readmissions. | | | | | | | | | |

(*Continued*)

**Table 4.** (*Continued*)

| | | OAS-CAHPS | CAHPS | CTM-3 | HCAHPS | Pres Ganey | |
|---|---|---|---|---|---|---|---|
| Schreiter et al., 2021 [15] | **Transitional follow-up support (telephone) after hospital discharge with possible remediation** plans and resources | Obs-C | | | | | +<br>x | Intervention got greater percentages of top-box scores than controls in 5 of 11 items: asking about having the needed help (100% vs. 93%, $p < .01$), educational materials (68% vs. 55% $p < .01$), understanding of responsibilities (69% vs. 59%, $p = .02$), instructions on whom to call with post-discharge questions (76% vs. 69%; $p = .04$), and global experience (57% vs. 46%, $p = .02$). But the readiness for and discharge satisfaction were similar. |
| | Telephone-based, continued follow-up care delivered by nurses after hospital discharge for: medication reconciliation, patient education about key symptoms that warrant contact with providers and ensuring that contact nurse information and follow-up appointments are in place, including remediation care (e.g., same-day clinic appointment, visit to the emergency room, or direct ward admission) if required. | | | | | | | |
| Thum et al., 2022 [31] | ***Teach back*** and new discharge summary, with workflow redesign | PPT | | | | + | +<br>x | Improved top-box percentages: Care Transitions, 52.4%–54.5% ($p < .001$); Discharge Information, 87.4%–90.1% ($p < .001$). Improved percentile rank 45.2–74.3 ($p = .0202$) for Discharge Information. The change in the percentiles for Care Transitions did not reach statistical significance: 56.8–65.8 ($p = .0591$). |
| | Nurses were trained in *teach back* intervention, their workflow was redesigned to accommodate the role, and a new discharge summary was created–linked to a *hard stop* in the electronic health record. | | | | | | | |
| March et al., 2022 [32] | **Pharmacist-led discharge education & follow-up on medication reconciliation** | PPT | | | | + | | Significant improvement in the top-box scores (52.6% *vs.* 67.3%; $p = < .001$) in the composite of medication-related HCAHPS results and its specific items: "tell you what the medicine was for" (67.7% *vs.* 81.9%; $p = .018$), "describe possible medicine side effects" (37.7% *vs.* 58.9%; $p = .004$), and "understood the purpose of taking medications" (52.3% *vs.* 63.7%; $p = .035$). |
| | Pharmacist-led medication reconciliation and education, sensitive to health literacy levels, prior and post discharge following alerts from the Electronic Medical Recording system. | | | | | | | |

OAS-CAHPS: Outpatient and Ambulatory Surgery Consumer Assessment of Healthcare Providers and System; CAHPS: Consumer Assessment of Healthcare Providers and System; CTM-3: Care Transitions Measure; HCAHPS: Hospital Consumer Assessment of Healthcare Providers and Systems; RCT: Randomized Controlled Trial; NR-CT: Non-Randomized Controlled Trials; PPT: Pre-Post Test study; Obs-C: Observational, comparative study.

RCTs) on enhanced QI support provided to PCHMs or other primary care practices. It was also found by eight studies on enhanced discharge support such as patient education, reengineered discharge, and/or telephone follow-up or transitional support. These latter studies, as a body of literature, had a substantive risk of bias. Finally, a transitional care intervention using a navigator was not effective at improving the patient experience for changes in informational support, although a subgroup analysis identified greater improvements for informational support among participants without health insurance.

Several risks of bias were identified in the reviewed literature across the study designs, limiting the internal and external validity of the findings. The studies on enhanced discharge support were typically single center. Even the multi-site studies on enhanced care coordination were performed within a network or organization. Hence, it is unknown how the findings apply elsewhere. For instance, the study with the lowest appraised risks of bias found that the more resource-intensive type of QI support provided to PCMH units was not superior to the dissemination of the QI tool alone for driving improvements in patient experience of care coordination activities. But that knowledge from a study within the VHA directly applies to

**Table 5. Transitional care interventions after inpatient care evaluated from the patient experience perspective as a primary outcome.**

| | Design & Analysis | Setting | Risk of Bias–within the study design: descriptive synthesis | |
|---|---|---|---|---|
| LaBedz et al 2022 [41] | Pragmatic RCT (patient-level randomization). Multivariable linear regression models, with a Bonferroni correction for the co-primary outcomes, sensitivity analyses for missing data imputation, and exploratory analyses for heterogeneity of treatment effects. | Minority-serving hospital with a transition to home; University of Illinois Hospital & Health Sciences System, Chicago, Illinois, USA | RCT (pragmatic) | Single site. Patients not masked to group allocation. Limited fidelity of the implementation: Only 29% Navigator group participants received the intervention per protocol. While the authors use a standardized measure, it is not a traditional patient experience measure (i.e., focus on social support dimensions partly addressed by the intervention)) and may not be specific enough or responsive enough for the intervention. |
| | **Intervention Type:** *Transitional Care* | | **Measure** | **Results—Descriptive Synthesis** |
| | The intervention group received an intervention during the index hospitalization and for 60 days post-discharge, which included 1) in-hospital visits by a community health worker to assess barriers to health/healthcare and to develop a personalized Discharge Patient Education Tool (DPET); 2) a post-discharge home visit by a community health worker to review the DPET; and 3) telephone-based peer coaching. | | PROMIS Informational Support (change score) x | No significant between-group differences in the 30-day change in informational support (adjusted difference: −0.01, 97.5% CI −2.0 to 1.9, p = 0.99), or any secondary outcomes such as emotional support [−0.12, 95% CI −1.5, 1.2, p = 0.86] or instrumental support [−0.43, 95% CI −1.7, 0.93, p = 0.53]. An exploratory subgroup analysis suggested greater improvements in 30-day informational support among Navigator group participants without health insurance (+11.9, 95% CI 2.3 to 21.4). |

**Keys:** Results per measure & metric **x** non-significant impact.

PROMIS, Patient-Reported Outcomes Measurement Information System; RCT: Randomized Controlled Trial.

the VHA. The VHA has implemented systematic QI programs, studies, or activities [42, 43], which may have contributed to building the capacity for effective QI activities across their units simply supported by an online QI tool, even without a centralized and more costly QI facilitation. The same may not apply to other healthcare organizations. Because context is a vital variable in QI science [44], future studies may consider generating evidence with validity across organizations and networks and their varied sets of circumstances. QI collaboratives or hubs, including those externally funded, might equitably foster the capacity to develop and assess QI activities across service delivery organizations [45].

Varied study designs were included in this review. While RCTs can produce a highest level of evidence when well conducted, a wider range of designs were included to reflect real-world improvement activities and evaluations, including pragmatic constraints [46]. Study limitations within each design were reflected in the appraised risks of bias, especially among the single-center discharge support studies. Moreover, there were pragmatic constraints in delivering some of the tested interventions. For example, a pharmacist-led patient education intervention was performed for patients discharged only during particular hours [32]. Similarly, a reengineered discharge support program was capped at a number of patients per day, as selected by nurses as "complex" cases [16]. These limitations may affect the fidelity of the improvement interventions and accuracy of the results and may reflect budget or other limitations (e.g., human resources, in-house improvement or research expertise) for the conduct or evaluation of QI activities [46]. Percentile ranks (i.e., provider ranks in patient experience performance relative to peers) were used as an additional metric only in one included paper (with a pre-post design), showing mixed results while the percentage of top-box scores had positive results [31]. Reporting results in more than one metric of a patient experience measure can be

important, especially for percentile ranks in pre-post studies. This is because the percentile rank metrics account for the performance of peer providers, it partly addresses the lack of a control group or the use of historical ones. Overall, the inclusion of varied study designs, interventions, and patient experience measures and metrics may inform further improvement activities and its study.

Across study designs, some common intervention approaches were found, such as enhanced supports for QI facilitation on care coordination [36, 37] or tailored approaches for at-risk populations [34, 35, 38, 40], often showing mixed patient experience results relative to standard supports. In turn, discharge support approaches that used a *reengineered discharge* or *teach back* strategies were also common but with less favorable results when compared to studies and reviews on other outcomes, such as reducing readmissions [47–49]. For example, one recent systematic review of *teach back* strategies applied to discharge education identified a 45% reduction in 30-day readmission [49]. For experience of care, studies using the *teach back* method had mixed results overall [28, 29, 31, 33] but neutral results in the RCTs [28, 29], either as the main strategy [28] or as part of a multi-modal strategy that also included a reengineered discharge, motivational interviewing, follow-up and transitional support, and overall language-concordant supports [29]. When found to be effective compared to a control intervention, the *teach back* method was used in isolation and specifically improved the patient experience with receiving information about new medicines [33]. For post-discharge support, a multi-modal intervention that included this component found neutral results [29], whereas the intervention on this component alone found positive results in five patient experience items [32]. Further research could compare the cost-effectiveness of multi-modal and single-method improvement approaches as well as determine whether multi-modal interventions, due to their complexity, are delivered with fidelity to the main active ingredients of the intervention [50, 51].

To be impactful on the patient experience, it may be key for studies to assess how patient-centered (e.g., how attentive, respectful, responsive, with genuine interest) was the delivery of a *teach back*, telephonic transitional support, and other supportive patient-provider communications intervention. This might include how tailored the communication was to the individual patient [18] to avoid the risks of being perceived as just another *box-ticking* exercise [52].

The only study (i.e., pragmatic trial) of a transitional support, specifically a navigator program, did not demonstrate increased effectiveness compared to usual care, although it may have had specific advantages for those who were uninsured [41]. Similar to other types of complex interventions, the quality and results of studies of transitional care interventions may benefit from improved implementation fidelity and intervention adaptability to emergent or context-specific patient needs and preferences–beyond a one-size-fits-all approach [41, 53, 54]. Additionally, tailored measures are needed to improve the formal assessment of patient experience with care that encompass transitional supports provided at home [55, 56].

All the included studies were conducted in the US, which may result from multiple factors. First, the CAHPS program in the US provides established, widely used standardized patient experience measures, which was a criterion for inclusion. Second, providers' performance on CAHPS measures is publicly reported and included in some value-based purchasing programs [11], which may have driven the interest of providers to develop and report these improvement activities. Third, CAHPS has been a primary quality target of a "person-centered" medical home, with the National Committee for Quality Assurance recommending administration of the CAHPS PCMH survey for PCMH transformation, and a related study found that CAHPS surveys were considered actionable for PCMH transformation [57]. Fourth, the VHA in the US has its own patient experience measurement system, which was used as a standardized assessment measure in studies externally reported and included here [35, 36]. Fifth,

improvement activities otherwise fitting the study eligibility criteria from other countries may have been conducted but not reported in English or through peer-reviewed journals.

In summary, the studies on the impact of improvement interventions related to care coordination, discharge support and care transitions on patient experience measures achieved mixed results, and the results are frequently not generalizable. To further advance this body of knowledge, activities might include conducting cross-organizational studies, using controlled designs, improving the implementation fidelity, and further integration of the patient experience perspective as a study outcome.

## Limitations

This review has several limitations. First, it excluded articles published in languages other than English or in the grey literature, which affects representativeness but reflects peer-reviewed knowledge readily available to an international audience. Second, representativeness is also affected by the exclusion of articles published before 2015. Third, for the data extraction, the second reviewer performed only confirmation tasks while two independent reviewer roles were used for all other tasks. Fourth, we did not perform a formal grading of the evidence or of the risks of bias within or across study types, within heterogenic health service delivery contexts, interventions, and study methods. Fifth, we did not include studies that used patient experience as a secondary study outcome. While including these studies may have added to the pool of evidence, it would have risked including studies in which improving the patient experience of care was not a primary goal.

## Conclusion

Enhanced support for improving care coordination and discharge support had mixed or neutral results for improving the patient experience with care beyond standard care or simpler supports. Substantial risk of bias and lack of comparison data between improvement approaches impede firmer conclusions, especially for the studies on enhanced discharge support, which were also typically single center. The multi-site studies on enhanced care coordination were also limited by being performed within an integrated delivery system, network, or organization, limiting the generalizability of the results across organizational and service delivery contexts. Finally, future studies should evaluate the impact of transitional care programs as assessed from the patient experience perspective, including with improved implementation fidelity and adaptive designs. There is a need to strengthen the body of evidence on the impact of health-service improvement activities assessed from the patient experience perspective, especially via enhanced patient discharge support and transitional approaches.

## Supporting information

**S1 Appendix. PRISMA 2020 checklist.**
(DOCX)

**S2 Appendix. Full search strategy per scientific database.**
(DOCX)

**S3 Appendix. Data extraction table.**
(XLSX)

**S4 Appendix. Quality appraisals.**
(DOCX)

## Author Contributions

**Conceptualization:** Tiago S. Jesus, Allen W. Heinemann, Anne Deutsch.

**Data curation:** Tiago S. Jesus, Brocha Z. Stern, Dongwook Lee, Manrui Zhang, Jan Struhar.

**Investigation:** Tiago S. Jesus, Brocha Z. Stern, Dongwook Lee, Manrui Zhang, Jan Struhar.

**Methodology:** Tiago S. Jesus, Brocha Z. Stern, Allen W. Heinemann, Neil Jordan, Anne Deutsch.

**Project administration:** Tiago S. Jesus.

**Supervision:** Allen W. Heinemann, Anne Deutsch.

**Validation:** Tiago S. Jesus, Brocha Z. Stern, Dongwook Lee, Manrui Zhang.

**Writing – original draft:** Tiago S. Jesus.

**Writing – review & editing:** Brocha Z. Stern, Allen W. Heinemann, Neil Jordan, Anne Deutsch.

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
