## [Decision Letter · Decision Letter 0]

27 Mar 2024

PONE-D-24-04801Systematic review of contemporary improvement interventions for care coordination, discharge support and transitional care from the patient experience perspectivePLOS ONE

Dear Dr. Jesus,

Thank you for submitting your manuscript to PLOS ONE. After careful consideration, we feel that it has merit but does not fully meet PLOS ONE’s publication criteria as it currently stands. Therefore, we invite you to submit a revised version of the manuscript that addresses the points raised during the review process.

We look forward to receiving your revised manuscript.

Kind regards,

Jingjing Qian

Academic Editor

PLOS ONE

Journal Requirements:

"Tiago S. Jesus completed part of this work and Manrui Zhang the totally of this work under a grant from the National Institute on Disability, Independent Living, and Rehabilitation Research (NIDILRR; 90ARHF0003). NIDILRR is a center within the Administration for Community Living (ACL), U.S. Department of Health and Human Services (HHS). The contents of this publication do not necessarily represent the policy of NIDILRR, ACL, or HHS, and the reader should not assume endorsement by the U.S. federal government. Dr Jesus initiated this work with the Northwestern University’s affiliation (under the grant support mentioned) and the work was completed with The Ohio State University’s affiliation."

Reviewers' comments:

Reviewer's Responses to Questions

**Comments to the Author**

1. Is the manuscript technically sound, and do the data support the conclusions?

Reviewer #1: Yes

Reviewer #2: Yes

2. Has the statistical analysis been performed appropriately and rigorously? 

Reviewer #1: Yes

Reviewer #2: Yes

3. Have the authors made all data underlying the findings in their manuscript fully available?

Reviewer #1: Yes

Reviewer #2: Yes

4. Is the manuscript presented in an intelligible fashion and written in standard English?

Reviewer #1: Yes

Reviewer #2: No

5. Review Comments to the Author

Reviewer #1: The authors reviewed contemporary literature describing the impact of care coordination, discharge support, and transitional care activities on patient experiences as assessed by formal patient-reported experience measures. The Reviewer thinks that this is a unique report and worthy.

The papers enrolled in this review were ultimately limited in number and varied in the type and degree of bias, which may have prevented them from drawing impactful conclusions. However, the Reviewer appreciates that it is significant in that it clarifies that such a situation exists because no systematic review focused on the impact that care coordination, discharge support, or transitional care activities have on the patient’s experience with care.

Reviewer #2: Thank you for the opportunity to review this work. I greatly appreciate the goal of synthesizing the transitional care literature. It's variability and breadth is difficult to navigate.

I believe the overarching theme of this manuscript is that there has not been much/enough work directly evaluating the patient experience related to transitions of care (TOC) services. The authors worked to capture the findings of a variable cross-section of the TOC literature specifically related to this outcome. This manuscript highlights a gap in the current literature and seems to be quite comprehensive. I believe that this is a valuable addition to the literature that would benefit from revision to enhance readability. Specific feedback can be found below:

Abstract:

- Is it possible to make the aim more concise? Can the domains of care coordination and discharge support be grouped together under a single heading (e.g. TOC)? This would bring the focal point to the process, in general, recognizing that there are many different features/components that this may encompass. This modification would likely also impact the title.

- Line 58: recommend a comma after "journals"

- Line 58: consider adding "published" between "studies" and "in"

- Line 58: consider adding "focused" before "on"

- Line 62: remove "finally"

- consider removing the number of RCT from the abstract or summarizing with the report of the 16 studies (e.g. "16 [10 RCT]"....

- consider removing the clarifications in parentheses from the abstract to improve readability. This would necessitate improving the phrasing for "enhanced supports" in line 64

- Lines 65-66: does "effectiveness" relate to the intervention or impact on patient experience? Please clarify.

- Line 70: it is unclear what the significance of "only for one patient experience item" means without the context of the tool used to assess the experience

- Line 71: "out of challenges with implementation fidelity" is unclear

- Line 72: It is unclear what differentiates "enhanced supports" from "simpler improvement approaches"

- Within the conclusion there is a need stated that is not represented by the rest of the abstract. In the absence of background information to demonstrate this need, it may make sense to state that the patient experience is important to the care process that there's a gap in the literature in this area.

Background

- Line 85: consider modifying "valuable"

- Line 86: please define "supports"

- Line 89: remove "to"

- Line 90: I don't think the examples within the parentheses are needed

- Line 96: remove "increasingly"

- Lines 101-102: remove "reflecting patient experience of care" since it was defined above

- Line 102: consider "calculating" instead of "computing"

- Line 104: consider whether "increasingly" is needed here

- Line 106: consider grouping care coordination and discharge support or making it clear that they are being linked to care transitions

- Line 108: please define "supported discharge"

- Line 109: add a comma after coordinators

- Lines 107-110: should there be a citation for the first sentence?

- Line 111: consider removing "of these"

- Line 112: the example within the parentheses is not needed if a similar example is provided in the next sentence

- Line 117: consider merging this paragraph with the previous paragraph

- Line 118: consider modifying to "on patient experiences with care"

- Line 120: consider modifying "synthesized as outcomes"

Methods

- Line 129: remove "the" after "to improve"

- Line 131: it may be more clear to add "on the patient experience" or something similar

- Line 137: it may be clearer to say, "full texts available, report patient experience as a primary outcome, have no more than two primary outcomes,...."

- Line 141: suggest removing the parentheses and stating, "assessments including...."

- Line 147: does "participants" mean participating locations or service providers? Please clarify.

- Line 150: please define "proxy respondents"

- Line 173: consider rephrasing (...searched in the time since be coming peer reviewed (2020)...."

- Line 181: consider modifying "involved" to something like "engaged" or "consulted"

- Line 183: consider substituting "including" for "such as"

- Line 192: citation for EPOC needed

- Line 194: citation needed for NHLBI tools

- Line 199: no comma needed after "design"

Results

- It would be helpful to have more information in the text about the patient experience measure used in each section

- Line 219: the information in the parentheses does not clarify the "supports" designation. What are "over and beyond models"?

- Lines 234-235: the description of the study is unclear, specifically, "...used QI facilitation (e.g. coaching) on the site's use of a...."

- Line 235: it may be clearer to remind the reader that the outcome that improved was patient experience

- Lines 235-236: more detail is needed to understand what makes up the "more resource-intensive arm" and what this means for the result

- Line 239: to clarify - are these care coordination items patient-experience outcomes?

- Lines 239, 250-251: the details of this study are unclear in the text ("encouraged the selection"). Does this refer to study design or outcomes.

- Line 268: write out "and"

- Line 287: consider including a short list of examples instead of the information in parentheses

- Line 287: consider removing "in turn"

- Line 289: remove the semi-colon and create two distinct sentences.

- Line 303: the information within the parentheses is redundant with the previous sentence

- Lines 300-308: this is the best description within the narrative. It is clear and concise.

Discussion

- Lines 319-320: some times the author speaks of two areas of review (care coordination and discharge support) and in some places three. I suggest being consistent throughout.

- Line 323: add a space between "usual" and "care"

Lines 324-325: Based on the previous sentence, should the findings of both the 7 and 8 studies be described in this sentence? Also, the third outcome (TOC) is left out of this paragraph.

- Lines 333-339: Is there a way to make this more general to the study as a whole instead of focusing on details of a single study previously provided within the results?

- Line 344: the last phrase of this sentence may be unnecessary

- Line 345: consider modifying "here" to something like "within this review."

- Line 345: consider updating the phrase "most solid evidence" and speak to rigor or level of evidence

- Line 354: consider removing "often"

- Line 356: the importance of the use of percentile ranks is unclear

- Line 359: clarify "in more than one metric of a patient experience"

- Line 363: consider removing "at varying levels of available resources"

- Line 366: consider removing information within the parentheses.

- Lines 371-379: this information sounds like it should go in the results section instead of the discussion

- Line 382: is "active ingredients" appropriate language within this study area?

- Lines 380-382: how does this relate to the assessment of patient experience?

- Lines 393-395: citation needed

- Lines 412-414: the second half of this sentence may not be appropriate for the limitations section

- Overall, I am missing a "take home point" from the discussion. It sounds like we don't really have enough data, need to focus studies on the patient experience, etc., but it would be helpful to have the large points summarized in one specific area.

Conclusion

- Line 427: the second of this half is not needed

- Lines 428-431: this seems redundant with previous sections and can be removed from conclusions

- Line 434: consider modifying "improvement" to "impact on" to remove the assumed directionality of the outcome. There is potential that some interventions may worsen the patient experience.

Figure 1

- Clear and easy to follow

- Consider adding the most common reasons for exclusion to the figure

Miscellaneous

- The summary tables are clear with appropriate details.

- It may be helpful to have a more lengthy description of the measures of patient experience.

6. PLOS authors have the option to publish the peer review history of their article (what does this mean?). If published, this will include your full peer review and any attached files.

Reviewer #1: No

Reviewer #2: **Yes: **Courtney E Gamston

---

## [Author Response · Author response to Decision Letter 0]

8 Apr 2024

The response follows in the file uploaded

---

## [Decision Letter · Decision Letter 1]

6 May 2024

Systematic review of contemporary interventions for improving discharge support and transitions of care from the patient experience perspective

PONE-D-24-04801R1

Dear Dr. Jesus,

We’re pleased to inform you that your manuscript has been judged scientifically suitable for publication and will be formally accepted for publication once it meets all outstanding technical requirements.

Kind regards,

Jingjing Qian

Academic Editor

PLOS ONE

Additional Editor Comments (optional):

Thank you for addressing reviewer's comments carefully. Your work will make a great impact in this field of transitions of care.

Reviewers' comments:

Reviewer's Responses to Questions

**Comments to the Author**

1. If the authors have adequately addressed your comments raised in a previous round of review and you feel that this manuscript is now acceptable for publication, you may indicate that here to bypass the “Comments to the Author” section, enter your conflict of interest statement in the “Confidential to Editor” section, and submit your "Accept" recommendation.

Reviewer #2: All comments have been addressed

2. Is the manuscript technically sound, and do the data support the conclusions?

Reviewer #2: Yes

3. Has the statistical analysis been performed appropriately and rigorously? 

Reviewer #2: Yes

4. Have the authors made all data underlying the findings in their manuscript fully available?

Reviewer #2: Yes

5. Is the manuscript presented in an intelligible fashion and written in standard English?

Reviewer #2: Yes

6. Review Comments to the Author

Reviewer #2: Thank you for the thoughtful consideration of the previous reviewer comments. This draft flows very smoothly for the reader. This literature with this field is not only extensive but difficult to synthesize due to its variability. This work does a great job of honing in on a specific opportunity for improvement while respecting that variability. I have just a couple small edits: 1) Abstract Aim: If you're an Oxford comma user, there should be a comma before "and" in line 55; 2) Materials and Methods, page 7, line 131, "PRISMA" is misspelled; 3) Results, p. 11 PRISMA diagram, full-text exclusion reasons, I suggest modifying "as no main target" to "not a primary outcome" or something similar and "students/trainees" to "student/trainee delivered."

Great work and thank you for this valuable work!

7. PLOS authors have the option to publish the peer review history of their article (what does this mean?). If published, this will include your full peer review and any attached files.

Reviewer #2: **Yes: **Courtney E. Gamston

---

## [Editor Report · Acceptance letter]

10 May 2024

PONE-D-24-04801R1 

PLOS ONE

Dear Dr. Jesus, 

I'm pleased to inform you that your manuscript has been deemed suitable for publication in PLOS ONE. Congratulations! Your manuscript is now being handed over to our production team.

Kind regards, 

on behalf of

Dr. Jingjing Qian 

Academic Editor

PLOS ONE